# QUASICONVEX SHALLOW NEURAL NETWORK

## ABSTRACT

Deep neural networks generally have highly non-convex structures, resulting in multiple local optima of network weights. The non-convex network is likely to fail, i.e., being trapped in bad local optima with large errors, especially when the task involves convexity (e.g., linearly separable classification). While convexity is essential in training neural networks, designing a convex network structure without strong assumptions (e.g., linearity) of activation or loss function is challenging. To extract and utilize convexity, this paper presents the QuasiConvex shallow Neural Network (QCNN) architecture with mild assumptions. We first decompose the network into building blocks where quasiconvexity is thoroughly studied. Then, we design additional layers to preserve quasiconvexity where such building blocks are integrated into general networks. The proposed QCNN, interpreted as a quasiconvex optimization problem, allows for efficient training with theoretical guarantees. Specifically, we construct equivalent convex feasibility problems to solve the quasiconvex optimization problem. Our theoretical results are verified via extensive experiments on common machine learning tasks. The quasiconvex structure in QCNN demonstrates even better learning ability than non-convex deep networks in some tasks.

## 1 INTRODUCTION

Neural networks have been at the heart of machine learning algorithms, covering a variety of applications. In neural networks, the optimal network weights are generally found by minimizing a supervised loss function using some form of stochastic gradient descent (SGD) (Saad (1998)), in which the gradient is evaluated using the backpropagation procedure (LeCun et al. (1998)). However, the loss function is generally highly non-convex, especially in deep neural networks, since the multiplication of weights between hidden layers and non-linear activation functions tend to break the convexity of the loss function. Therefore, there are many local optima solutions of network weights (Choromanska et al. (2015)). While some experiments show that certain local optima are equivalent and yield similar learning performance, the network is likely to be trapped in bad local optima with a large loss.

**Issue 1:** Is non-convex deep neural networks always better?

Deep neural networks have shown success in many machine learning applications, such as image classification, speech recognition, and natural language processing (Hinton & Salakhutdinov (2006); Ciregan et al. (2012), Hinton et al. (2012), and Kingma et al. (2014)). Many people believe that the multiple layers in deep neural networks allow models to learn more complex features and perform more intensive computational tasks. However, deep neural networks are generally highly non-convex in the loss function, which makes the training burdensome. Since the loss function has many critical points, which include spurious local optima and saddle points (Choromanska et al. (2015)), it hinders the network from finding the global optima and makes the training sensitive to the initial guess. In fact, (Sun et al. (2016)) pointed out that increasing depth in neural networks is not always good since there is a trade-off between non-convex structure and representation power. In some engineering tasks requiring additional physical modeling, simply applying deep neural networks is likely to fail. Even worse, we usually don't know how to improve the deep neural networks during a failure since it is a black box procedure without many theoretical guarantees.

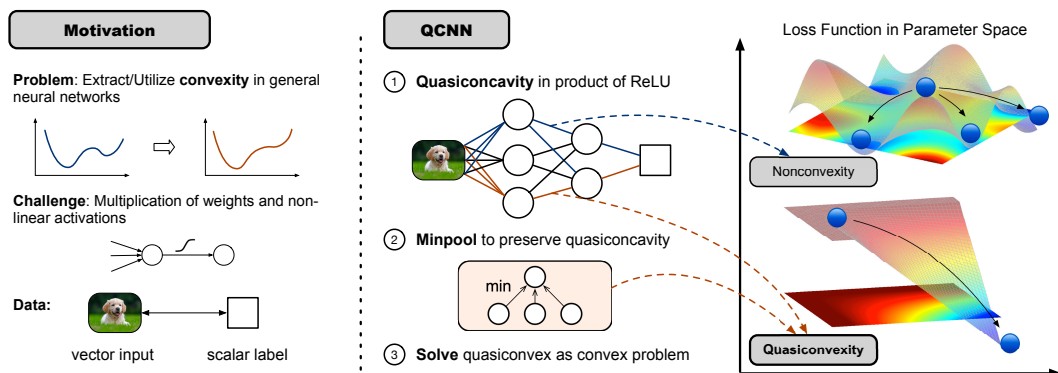

Figure 1: **Proposed Method.** (Left) The motivation and challenge of this study. (Middle) We design a quasiconvex neural network structure to efficiently train for optimal network weights in a quasiconvex optimization problem. The quasiconvexity is studied and preserved via special pooling layers. (Right) Unlike non-convex loss function, the quasiconvex loss function of our design allows for finding the global optima.

**Issue 2:** Solution to non-convexity is not practical.

To overcome non-convexity in neural networks, new designs of network structure were proposed. The first line of research focused on specific activation functions (e.g., linear or quadratic) and specific target functions (e.g., polynomials) (Andoni et al. (2014)) where the network structure can be convexity. However, such methods were limited in practical applications (Janzamin et al. (2015)). Another line of research aimed at deriving the dual problem of the optimization problem formulated by neural network training. Unlike the non-convex neural network, its dual problem is usually convex. Then, conditions ensuring strong duality (zero duality gap and dual problem solvable) were discussed to find the optimal solution to the neural network. For example, Ergen & Pilanci (2020) derived the dual problem for neural networks with ReLU activation, and Wang et al. (2021) showed that parallel deep neural networks have zero duality gap. However, the derivation of strong duality in the literature requires the planted model assumption, which is impractical in many real-world datasets. Aside from studying the convexity in network weights, some work explored the convexity in data input and label. For instance, an input convex structure with given weights Amos et al. (2017) altered the neural network output to be a convex function of (some of) the inputs. Nevertheless, such a function is only an inference procedure with given network weights.

In this work, we introduce QCNN, the first **Q**uasi**C**onvex shallow **N**eural **N**etwork structure that learns the optimal weights in a quasiconvex optimization problem. We first decompose a general neural network (shown in the middle of Figure 1) into building blocks (denoted by distinct colors). In each building block, the multiplication of two weights, as well as the non-linear activation function in the forward propagation, makes the building block non-convex. Nevertheless, inspired by Boyd et al. (2004), we notice that the multiplication itself is quasiconcave if the activation function is ReLU. The quasiconvexity (quasiconcavity) is a generalization of convexity (concavity), which shares similar properties, and hence, is a desired property in the neural network. To preserve quasiconcavity in the network structure when each building block is integrated, we design special layers (e.g., minimization pooling layer), as shown in the middle of Figure 1. In doing so, we arrive at a quasiconvex optimization problem of training the network, which can be equivalently solved by tackling convex feasibility problems. Unlike non-convex deep neural networks, the quasi-convexity in QCNN enables us to learn the optimal network weights efficiently with guaranteed performance.

## 2 RELATED WORK

**Failure of training non-convex neural networks**. In training a non-convex neural network, the commonly used method, such as gradient descent in the backpropagation procedure, can get stuck in bad local optima and experience arbitrarily slow convergence (Janzamin et al. (2015)). Explicit examples of the failure of network training and the presence of bad local optima have been discussed in (Brady et al. (1989); Frasconi et al. (1993)). For instance, Brady et al. (1989) constructed simple cases of linearly separable classes that backpropagation fails. Under non-linear separability setting, Gori & Tesi (1992) also showed failure of backpropagation. These studies indicate that deep neural

networks are not suitable for all tasks. Therefore, it motivates us to think: can simple networks with convex structure beat deep networks with non-convex structure in some tasks?

**Convexity in neural network.** The lack of convexity has been seen as one of the major issues of deep neural networks (Bengio et al. (2005)), drawing much research in the machine learning community. Many people studied convex structures and convex problems in neural networks. For instance, (Bengio et al. (2005); Andoni et al. (2014); Choromanska et al. (2015); Milne (2019); Rister & Rubin (2017)) showed that training a neural network under some strong conditions can be viewed as a convex optimization problem. (Farnia & Tse (2018); Ergen & Pilanci (2020); Wang et al. (2021); Pilanci & Ergen (2020)) studied convex dual problems of the neural network optimization and derived strong duality under specific assumptions.

Aside from directly studying convexity in neural networks, people also discussed conditions where the local optima become global. For example, Haeffele & Vidal (2015) presented that if the network is over-parameterized (i.e., has sufficient neurons) such that there exist local optima where some of the neurons have zero contribution, then such local optima is global optima (Janzamin et al. (2015)). Similarly, Haeffele & Vidal (2017) also showed that all critical points are either global minimizers or saddle points if the network size is large enough. However, such studies only provide theoretical possibilities while efficient algorithms to solve (infinitely) large networks are missing. Based on the existing literature, we restrict our research to finding a simple but practical neural network with some convexity to provide performance guarantees.

**Quasiconvex optimization problem.** The study of quasiconvex functions, as well as quasiconvex optimization problems, started from (Fenchel & Blackett (1953); Luenberger (1968)) and has become popular nowadays since the real-world function is not always convex. For instance, quasiconvex functions have been of particular interest in economics (Agrawal & Boyd (2020)), modeling the utility functions in an equilibrium study (Arrow & Debreu (1954); Guerraggio & Molho (2004)). The quasiconvex optimization has been applied to many applications recently, including engineering (Bullo & Liberzon (2006)), model order reduction (Sou), computer vision (Ke & Kanade (2007)) and machine learning (Hazan et al. (2015)). Among many solutions to quasiconvex optimization problems, a simple algorithm is bisection (Boyd et al. (2004)), which solves equivalent convex feasibility problems iteratively until converging.

## 3 PRELIMINARY

To model for the general case, we consider a $L$-layer network with layer weights $\mathbf{W}_l \in \mathbb{R}^{m_{l-1} \times m_l}$, $\forall l \in [L]$, where $m_0 = d$ and $m_L = 1$ are the input and output dimensions, respectively. As the dimensions suggest, the input data is a vector, and the output is a scalar. Given a labeled dataset $\mathcal{D} = \{(\mathbf{x}_i, \mathbf{y}_i)\}_{i=1}^n$ with $n$ samples, we consider a neural network with the following architecture.

$$f_\theta(\mathbf{X}) = \mathbf{h}_L, \quad \mathbf{h}_l = g(\mathbf{h}_{l-1}\mathbf{W}_l), \quad \forall l \in [L] \tag{1}$$

where $\mathbf{h}_l$ denotes the layer activation and $\mathbf{h}_0 = \mathbf{X} \in \mathbb{R}^{n \times d}$ is the data matrix. Here, $\theta = \{\mathbf{W}_l\}_{l=1}^L$ are the network weights which need to be optimized via training, and $g(\cdot)$ is the non-linear activation function. The network is trained with $L^2$ loss as follows:

$$\hat{\theta} = \arg\min_\theta \frac{1}{2}\|f_\theta(\mathbf{X}) - \mathbf{y}\|_2^2, \tag{2}$$

where $\mathbf{y} \in \mathbb{R}^n$ is the data label vector. The loss function in Equation 2 is generally non-convex because of the multiplication of weights as well as non-linear activation functions. As discussed previously, non-convexity will likely cause the network to be trapped in a bad local optima with large errors. Therefore, we still want to extract some convexity in this loss function to help with the training process. In this paper, we will show that quasiconvexity and quasiconcavity are hidden in the network. These properties can be utilized to construct a convex optimization problem to train the optimal network weights. Here, we introduce the definition of quasiconvexity and quasiconcavity.

**Definition 1.** *A function $f : \mathbb{R}^d \to \mathbb{R}$ is quasiconvex if its domain and all its sublevel sets $\{x \in \mathbf{dom}\, f | f(x) \leq \alpha\}, \forall \alpha$ are convex. Similarly, a function is quasiconcave if $-f$ is quasiconvex, i.e., every superlevel set $\{x | f(x) \geq \alpha\}$ is convex.*

We also note, since convex functions always have convex sublevel sets, they are naturally quasiconvex, while the converse is not true. Therefore, the quasiconvexity can be regarded as a generalization of convexity, which is exactly what we seek in the non-convex deep neural networks.

## 4  QUASI-CONCAVE STRUCTURE

For designing a quasiconvex structure of neural networks, we start by considering the simplest and most representative building block in a network and analyze its characteristics. Specifically, we consider the network

$$f(\mathbf{w}_1; w_2) = g(g(\mathbf{x}^\top \mathbf{w}_1)w_2), \tag{3}$$

where $\mathbf{x} \in \mathbb{R}^d$ is the input data, $\mathbf{w}_1 \in \mathbb{R}^d$ and $w_2 \in \mathbb{R}$ are the weights for two hidden layers. We analyze such a two-layer structure because it is the simplest case in neural networks yet can be generalized to deep neural networks. In Equation 3, the network is not convex in weights $(\mathbf{w}_1; w_2)$ because (1) the network function contains the multiplication of the weights and (2) the activation function $g(\cdot)$ is usually non-linear.

Nevertheless, we still want to explore the potential possibilities of the network becoming convex or being related to convex. Inspired by Boyd et al. (2004), we notice that although the multiplication of two weights in the forward propagation makes the network non-convex, the multiplication itself is quasiconcave in specific circumstances. For example, the product of two variables forms the shape of a saddle, which is not convex in those variables. However, if we restrict these two variables to be positive, the saddle shape will reduce to a quasiconcave surface, as shown in Figure 2.

**Lemma 1.** *The function $f(w_1, w_2) = w_1 w_2$ with $\mathbf{dom}\ f = \mathbb{R}_+^2$ is quasiconcave.*

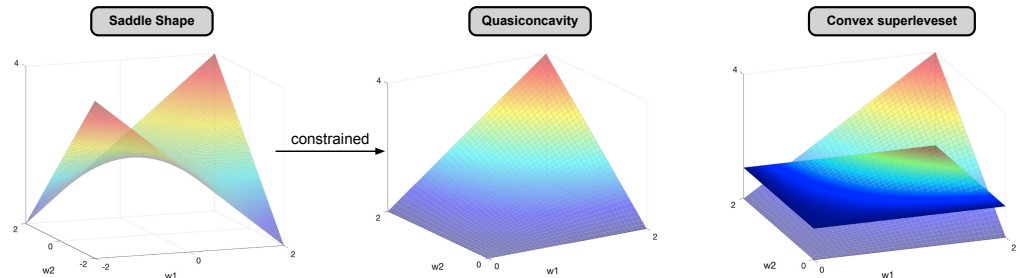

Figure 2: (Left) The function $f(w_1, w_2) = w_1 w_2$ has a saddle shape. (Middle) Constrained on positive domain, i.e., $\mathbf{dom}\ f = \mathbb{R}_+^2$, the function becomes quasiconcave. (Right) The quasiconcave function always has convex superlevel sets.

Motivated by Lemma 1 and Figure 2, to preserve the property of quasiconcavity of the network in Equation 3, a straightforward approach is to assume the network weights $(\mathbf{w}_1; w_2)$ to be non-negative, like in (Amos et al. (2017)). However, this assumption will significantly reduce the neural network's representation power. In fact, suppose there are $m$ weights, constraining all the weights to be non-negative will result in only $1/2^m$ representation power. To bypass this impractical assumption, we notice that some activation functions naturally restrict the output to be non-negative. For example, the ReLU activation function $g(\mathbf{x}) = \max\{0, \mathbf{x}\}$ forces the negative input to be zero. Therefore, we can demonstrate that the network in Equation 3 with ReLU activation function is quasiconcave in the network weights, as shown in Theorem 1.

**Theorem 1.** *The neural network in Equation 3 with ReLU activation function $g(\cdot)$ is quasiconcave in the network weights $(\mathbf{w}_1; w_2)$.*

*Proof.* To prove quasiconcavity, we need to show that all superlevel sets

$$S_\alpha = \{(\mathbf{w}_1; w_2) | f(\mathbf{w}_1; w_2) \geq \alpha\}, \quad \forall \alpha \in \mathbb{R}$$

are convex sets. When $\alpha \leq 0$, the superlevel set is the complete set, i.e., $S_\alpha = \mathbf{dom}\ f$ due to the ReLU activation function. Hence, $S_\alpha$ is evidently convex. When $\alpha > 0$, the superlevel set is neither the empty set nor the complete set. For any two elements $(\hat{\mathbf{w}}_1; \hat{w}_2), (\tilde{\mathbf{w}}_1; \tilde{w}_2) \in S_\alpha$, we aim to show that $(\lambda\hat{\mathbf{w}}_1 + (1 - \lambda)\tilde{\mathbf{w}}_1; \lambda\hat{w}_2 + (1 - \lambda)\tilde{w}_2) \in S_\alpha$ for $\lambda \in (0, 1)$. From the condition $\alpha > 0$, we

know that $\mathbf{x}^\top \hat{\mathbf{w}}_1 > 0$ and $\hat{w}_2 > 0$, as well as $\mathbf{x}^\top \tilde{\mathbf{w}}_1 > 0$ and $\tilde{w}_2 > 0$. Therefore, we would have

$$
\begin{aligned}
f(\lambda\hat{\mathbf{w}}_1 + (1-\lambda)\tilde{\mathbf{w}}_1; \lambda\hat{w}_2 + (1-\lambda)\tilde{w}_2) &= \left[\lambda\mathbf{x}^\top\hat{\mathbf{w}}_1 + (1-\lambda)\mathbf{x}^\top\tilde{\mathbf{w}}_1\right]\left[\lambda\hat{w}_2 + (1-\lambda)\tilde{w}_2\right] \\
&= \lambda^2\mathbf{x}^\top\hat{\mathbf{w}}_1\hat{w}_2 + (1-\lambda)^2\mathbf{x}^\top\tilde{\mathbf{w}}_1\tilde{w}_2 + \lambda(1-\lambda)\left[\mathbf{x}^\top\hat{\mathbf{w}}_1\tilde{w}_2 + \mathbf{x}^\top\tilde{\mathbf{w}}_1\hat{w}_2\right] \\
&\geq \lambda^2\alpha + (1-\lambda)^2\alpha + \lambda(1-\lambda)\left[\mathbf{x}^\top\hat{\mathbf{w}}_1\tilde{w}_2 + \mathbf{x}^\top\tilde{\mathbf{w}}_1\hat{w}_2\right] \\
&\geq \lambda^2\alpha + (1-\lambda)^2\alpha + \lambda(1-\lambda)\left[\frac{\alpha}{\hat{w}_2}\tilde{w}_2 + \frac{\alpha}{\tilde{w}_2}\hat{w}_2\right] \\
&\geq \lambda^2\alpha + (1-\lambda)^2\alpha + \lambda(1-\lambda) \times 2\alpha = \alpha[\lambda^2 + (1-\lambda)^2 + 2\lambda(1-\lambda)] = \alpha.
\end{aligned}
$$

$\square$

In Theorem 1, we show that the simple two-layer network in Equation 3 is quasiconcave in network weights, given that the activation function is ReLU. Then, a natural question arises: does this property of quasiconcavity remain in deeper networks? Unfortunately, the quasiconcavity does not hold in more complex neural networks due to one fact: the summation of quasiconvex (quasiconcave) functions is not necessarily quasiconvex (quasiconcave). The deeper networks can be regarded as weighted summations of many networks in Equation 3, hence, not quasiconcave anymore. Therefore, we aim to design new network structures to preserve the property of quasiconcavity to more general neural networks.

To achieve this goal, we focus on the operations that preserve quasiconcavity, including (1) the composition of a non-decreasing convex function, (2) the non-negative weighted minimization, and (3) the supremum over some variables. Among these operations, we choose the minimization procedure because it is easy to apply and has a simple gradient. Specifically, we can apply a minimization pooling layer to integrate the simple networks in Equation 3, as shown in Figure 3. In doing so, we manage to extend the network in the simplest building block to more general structures, where quasiconcavity is ensured by Lemma 2 and visually explained in Figure 4. Meanwhile, we note that the proposed network is still a shallow network. Although infinitely stacking layers with appropriate minimization pooling layers can also keep the entire network convex, too many minimization pooling layers will damage the representation power of the neural network.

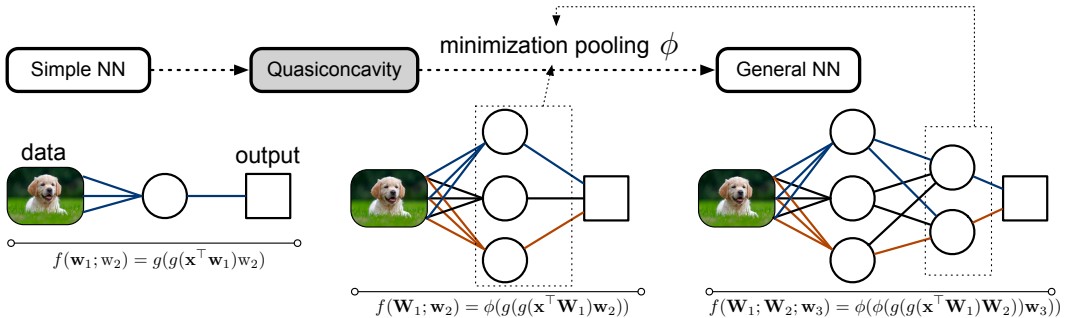

Figure 3: The structure of quasiconvex shallow neural network (QCNN).

**Lemma 2.** *Provided that $f_1, \cdots, f_n$ are quasiconcave functions defined on the same domain, the non-negative weighted minimum*

$$
f := \min\{a_1 f_1, a_2 f_2, \cdots, a_n f_n\}
$$

*is quasiconcave given $a_1, \cdots, a_n \in \mathbb{R}_+$.*

*Proof.* The superlevel set $S_\alpha = \{\mathbf{x} \in \mathbf{dom}\, f | f(\mathbf{x}) \geq \alpha\}$ of $f$ can be regarded as:

$$
S_\alpha = \{\mathbf{x} \in \mathbf{dom}\, f | \min\{a_1 f_1(\mathbf{x}), a_2 f_2(\mathbf{x}), \cdots, a_n f_n(\mathbf{x})\} \geq \alpha\} = \cap_{i=1}^n \{\mathbf{x} \in \mathbf{dom}\, f | f_i \geq \frac{\alpha}{a_i}\},
$$

which is the intersection of (convex) superlevel sets of $f_i (i = 1, \cdots, n)$.  $\square$

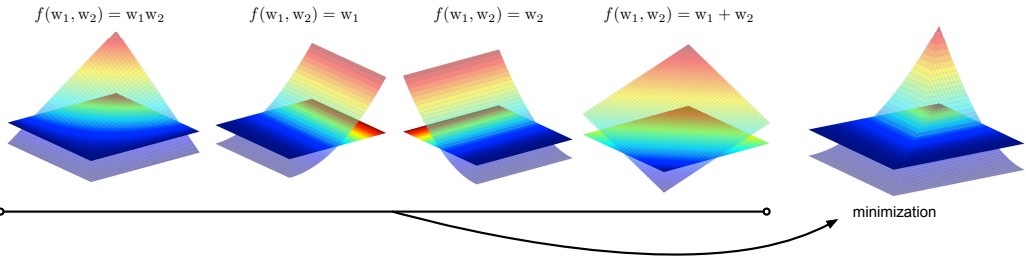

Figure 4: The minimization of quasiconcave functions on the same domain is still quasiconcave.

## 5 QUASICONVEX OPTIMIZATION OF NEURAL NETWORK

In Section 4, we design a neural network structure where output $f(\theta)$ is a quasiconcave function over the network weights $\theta$. To further utilize the property of quasiconcavity, in this section, we propose to train the neural network as a quasiconvex optimization problem. Even though function $f(\theta)$ is quasiconcave, the optimization problem in Equation 2 is not quasiconvex, since the $L^2$ loss is not monotonic. However, if we restrict the network output to be smaller than the network labels, i.e., $f(\theta) \leq y$, the $L^2$ loss is non-increasing in this range. Therefore, the resulting loss function in Equation 2, as a composition of a convex non-increasing function over a quasiconcave function, is quasiconvex. That is, the training of QCNN is an unconstrained quasiconvex optimization problem

$$P^* = \min_\theta l(\theta) = \min_\theta \frac{1}{2}\|f(\theta) - y\|_2^2. \tag{4}$$

To solve the quasiconvex optimization problem in Equation 4, we can transform it into an equivalent convex feasibility problem. Let $\varphi_t(\theta) := y - t - f(\theta), t \in \mathbb{R}$ be a family of convex functions satisfying $l(\theta) \leq t \iff \varphi_t(\theta) \leq 0$. Then, the quasiconvex optimization problem in Equation 4 can be equivalently considered as

$$\min_\theta \quad l(\theta) \qquad \Longleftrightarrow \qquad \min_{\theta,t} \quad t \qquad \Longrightarrow \qquad \text{find} \quad \theta \tag{5}$$
$$s.t. \quad l(\theta) \leq t \qquad\qquad s.t. \quad \varphi_t(\theta) \leq 0.$$

The problem in Equation 5 is a convex feasibility problem since the inequality constraint function is convex. For every given value $t$, we can solve the convex feasibility problem. If the convex feasibility problem is feasible, i.e., $\exists \theta, \varphi_t(\theta) \leq 0$, this point $\theta$ is also feasible for the quasiconvex problem by satisfying $l(\theta) \leq t$. It indicates that the optimal value $P^*$ is smaller than $t$, i.e., $P^* \leq t$. In this circumstance, we can reduce the value $t$ and conduct the above procedure again to approach the optimal value $P^*$. On the other hand, if the convex feasibility problem is infeasible, we know that $P^* \geq t$. In this case, we should increase the value of $t$. Through this procedure, the quasiconvex optimization problem in Equation 4 can be solved using bisection, i.e., solving a convex feasibility problem at each step (Boyd et al. (2004)). The procedure is summarized as Algorithm 1.

---

**Algorithm 1** QCNN Training Process

---

1: **given** $l \leq P^*$, $u \geq P^*$, tolerance $\epsilon > 0$      ▷ lower/upper bounds of optimal value
2: **while** $u - l > \epsilon$ **do**      ▷ convergence criterion
3:      $t := (l + u)/2$
4:      Solve the convex feasibility problem in Equation 5
5:      **if** problem Equation 5 is feasible **then**
6:          $u := t$      ▷ record the feasible point $\theta$
7:      **else**
8:          $l := t$
9:      **end if**
10: **end while**      ▷ return the current feasible point $\theta$

---

**Remark 1.** *The quasiconvex optimization problem in Equation 4 has zero duality gap. The proof can be derived by verifying that our unconstrained quasiconvex optimization problem in Equation 4 satisfies the condition in Fang et al. (2014). Therefore, the quasiconvex optimization problem could also be solved via exploring its dual problem.*

## 6 EXPERIMENTS

We use the proposed framework in Section 5 to conduct several machine learning tasks, comparing QCNN to deep neural networks. Our experiments aim to validate the core benefits of QCNN: (1) the convexity, even in shallow networks, makes learning more accurate than non-convex deep networks in some tasks, and (2) the convexity enables the network to be more robust and converge faster.

### 6.1 FUNCTION APPROXIMATION

Since the purpose of neural networks can be generally seen as learning a mapping from input $x$ to label $y$, in this section, we evaluate the performance of using QCNN to approximate some function.

**Synthetic dataset.** For synthetic scenario, the dataset is generated by randomly sampling $x$ from a uniform distribution $\text{Unif}(-1, 1)$ and calculating the corresponding label $y = f(x)$ given function $f$. We generate 1,000 samples for training and 200 samples for testing, where the mean square error (MSE) of the testing set is used to evaluate the model performance.

The results of approximating various functions are summarized in Figure 5. As we see, the performance of deep neural networks depends on the choice of initial guess of network weights. In the first two experiments (first two rows in Figure 5), the deep network seems to be trapped in a bad local optima, which corresponds to a relatively large MSE. In the third experiment, the deep network arrives at a good local optima. However, it still exhibits certain flaws at the non-differentiable points (turning points) in the function $f$. It matches the finding of Brady et al. (1989) where deep neural networks fail in simple cases of linearly separable classifying tasks. On the contrary, although QCNN uses a shallow structure, its quasiconvexity nature enables it to learn piecewise linear functions to approximate function $f$. In many replications of experiments, we find that learning procedure of QCNN is more robust to initial guess of network weights since it is quasiconvex. Moreover, QCNN demonstrates a quicker convergence when learning the function $f$.

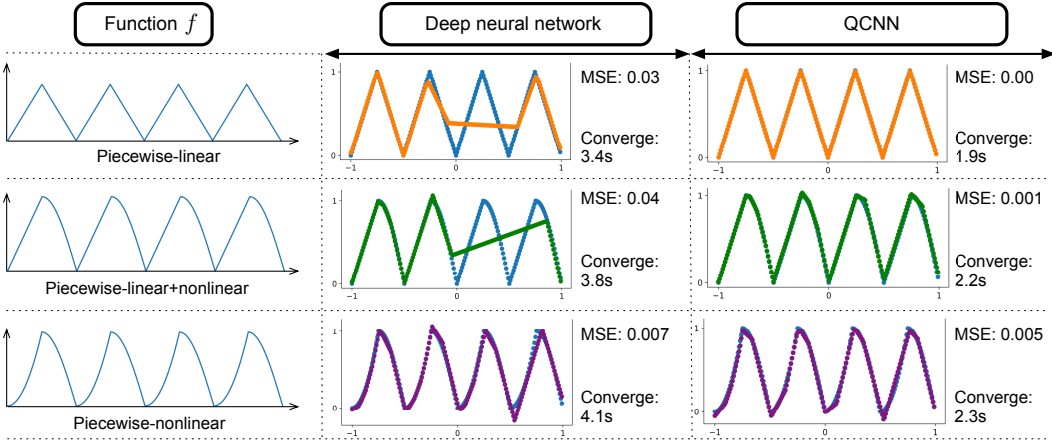

Figure 5: The performance of approximating functions using deep neural networks and QCNN: QCNN tends to behave better because the quasiconvex structure enables it to learn piecewise linear mappings more efficiently.

**Contour detection dataset**. For the real-world application of function approximation, we consider the task of detecting the contour of an object, which is usually the first step for many applications in computer vision, such as image-foreground extraction (Banerjee et al. (2016)), simple-image segmentation (Saini & Arora (2014)), detection, and recognition (Kulkarni et al. (2013)). The experiment is conducted on the Berkeley Segmentation Dataset (Arbeláez et al. (2013)) composed of 200 training, 200 testing, and 100 validation images. Each image has between five and six manually annotated labels representing the ground truth (Yang et al. (2019)). These labels are also used to calculate two metrics: optimal dataset scale (ODS) and optimal image scale (OIS) to evaluate the model performance. The comparison was performed against DeepNet (Kivinen et al. (2014)).

In the experiment, we find that the performance of QCNN and DeepNet depend on the objects in the image. For some objects with clear and angular contours, e.g., a phone in Figure 6 (a), detecting such a contour can be seen as learning a closed polygon with piecewise linear functions defining its edges. For such a class of images, the ODS of QCNN achieves $0.824$ compared to $0.784$ of DeepNet, while the OIS of QCNN achieves $0.831$ compared to $0.798$ of DeepNet. On the contrary, DeepNet has better accuracy in recognizing complex (e.g., highly non-linear) contours. In this class of images, the ODS of QCNN is $0.717$ compared to $0.743$ of DeepNet, while and the OIS of QCNN is $0.729$ compared to $0.760$ of DeepNet. To conclude, QCNN with the quasiconvex structure still outperforms the deep networks when the task involves some characteristics related to convexity.

**Mass-damper system dataset**. Aside from synthetic functions and irregular functions, we also learn functions that have physical meanings. Specifically, we consider the mass-damper system, which can be depicted as: $\dot{\boldsymbol{q}} = -\boldsymbol{D}\boldsymbol{R}\boldsymbol{D}^\top\boldsymbol{M}^{-1}\boldsymbol{q}$. In the system, $\dot{\boldsymbol{q}}$ is a vector of momenta, $\boldsymbol{D}$ is the incidence matrix of the system, $\boldsymbol{R}$ is the diagonal matrix of the damping coefficients for each line of the system, and $\boldsymbol{M}$ is the diagonal matrix of each node mass of the system. Thus, we can set $\mathbf{y} = \dot{\boldsymbol{q}}$ and $\mathbf{x} = \boldsymbol{q}$ with the goal to learn the parameter matrix $-\boldsymbol{D}\boldsymbol{R}\boldsymbol{D}^\top\boldsymbol{M}^{-1}$. We simulate the dataset for a 10-node system and obtain 6,000 samples for 1-min simulations with a step size to be 0.01s (Li et al. (2022)). Figure 6 (b) shows the prediction error (MSE) during training epochs, where QCNN converges faster to a smaller error, compared to deep neural networks. This is perhaps because that the target parameter matrix in the system constructs a linear bridge between input $\mathbf{x}$ and label $\mathbf{y}$.

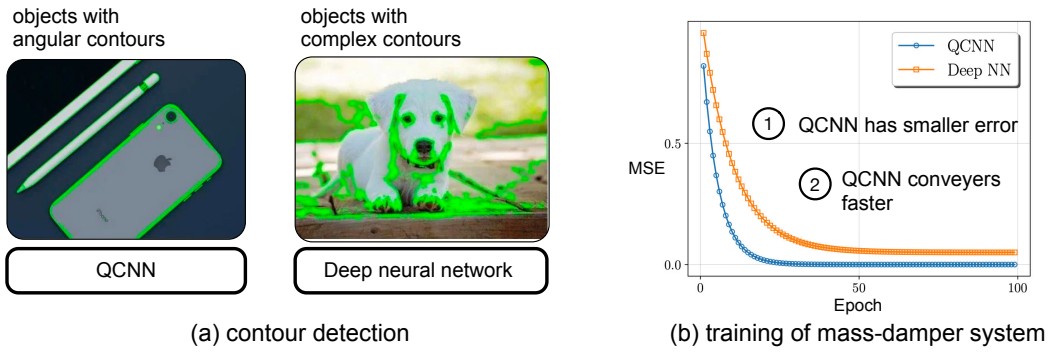

(a) contour detection · (b) training of mass-damper system

Figure 6: (a) QCNN works better in detecting angular contours while deep networks are better for detecting complex contours. (b) MSE against training epochs in learning the mass-damper function.

## 6.2 CLASSIFICATION TASK

The experiments in Section 6.1 represent the regression tasks. In this section, we further consider the classification task, covering two major categories of machine learning applications.

**Change point detection of distributions**. Finding the transition of the underlying distribution of a sequence has various applications in engineering fields, such as video surveillance (Sultani et al. (2018)), sensor networks Xie & Siegmund (2013) and infrastructure health monitoring Liao et al. (2019). Aside from engineering tasks, it is also important in many machine learning tasks, including speech recognition Chowdhury et al. (2012), sequence classification Ahad & Davenport (2020), and dataset shift diagnosis Lu et al. (2016). To simulate a sequence of measurements, we randomly generate the pre-change sequence from normal distribution $\mathcal{N}(0, 0.2)$ and generate the post-change sequence from $\mathcal{N}(1, 0.1)$ where the change time is $\lambda = 50$. The time-series sequence is shown in the left part of Figure 7.

Using the neural network to detect the change point $\lambda$ can be seen as classifying pre-change data and post-change data using five samples in a shifted window. The classifying threshold is chosen as $\alpha$, i.e., the maximum false alarm rate (Liao et al. (2016)). The results are shown in Figure 7 using 1,000 Monte Carlo experiments. As we see, QCNN shows a smaller average detection delay than deep neural networks. Meanwhile, it seems that QCNN is less likely to falsely report a fake change, since the empirical false alarm rate of QCNN is below that of deep networks, and is mostly below the theoretical upper bound $\alpha$ (especially when $\alpha \to 0$). QCNN outperforms the deep neural network in this task because the transition of the distribution is abrupt, as shown in Figure 7 (Left). The abrupt change results in a non-differentiable/non-smooth point in the mapping to be learned, which is more efficiently represented by QCNN via piecewise linear functions.

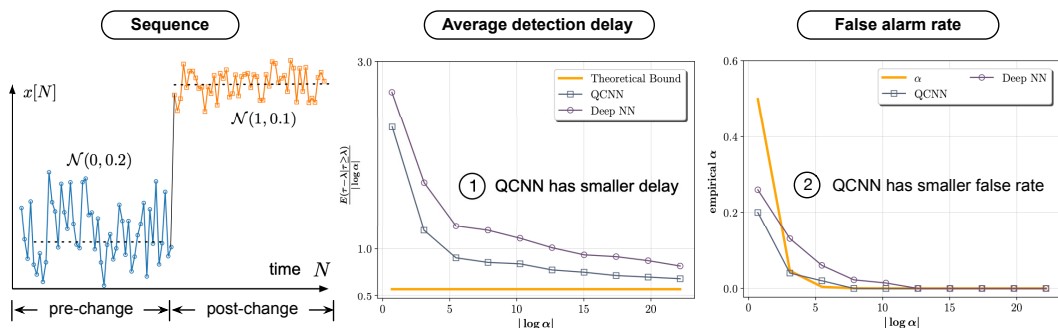

Figure 7: (Left) The change of underlying distribution of the sequence. (Middle) The average detection delay of detecting the distribution change using QCNN and deep neural networks. (Right) The false alarm rate of detecting the distribution change using QCNN and deep neural networks.

**Solar meters classification**. The UMass Smart dataset (Laboratory for Advanced System Software (Accessed Sep. 2022.)) contains approximately 600,000 meters from a U.S. city with a one-hour interval between each meter reading (Cook et al. (2021)). Among these meters, around 1,973 have installed solar panels, and are labeled as solar in the classification task. The remainder of the meters are labeled as non-solar. The average smart meter readings, including the household electricity consumption and the PV generation, as well as the household's address, are considered as input data to classify whether a meter has solar panels. We randomly select 20,000 samples from this dataset to train and select 1,000 samples to test the performance. Figure 8 shows the location of all the meters and solar meters. As we see, the meters that have solar installed are concentrated in a roughly convex area instead of being spread over in the entire area. Therefore, learning the classifier for solar meters using the feature of address is equivalent to learning a convex domain. It could explain that the classification accuracy of QCNN (94.2%) outperforms that of deep networks (92.7%).

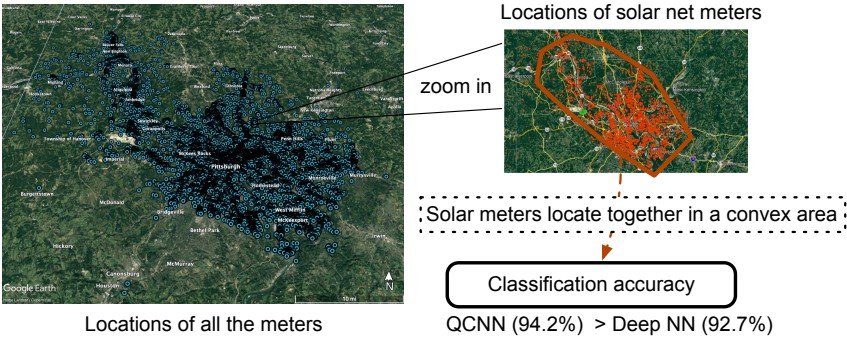

Figure 8: The locations of (solar) meters and the classification accuracy of using QCNN and deep neural network to classify the solar meters.

## 7 CONCLUSION

In this work, we analyze the problem of convex neural networks. First, we observe that deep neural networks are not suitable for all tasks since the network is highly non-convex. The non-convex network could fail, i.e., being trapped in bad local optima with large errors, especially when the task involves convexity (e.g., linearly separable classification). Therefore, it motivates us to design a convex structure of neural networks to ensure efficient training with performance guarantees. While convexity is damaged due to the multiplication of weights as well as non-linear activation functions, we manage to decompose the neural network into building blocks, where the quasiconvexity is thoroughly studied. In the building block, we find that the multiplication of ReLU output is quasiconcave over network weights. To preserve the property of quasiconcavity when such building blocks are integrated into a general network, we design minimization pooling layers. The proposed Quasi-Convex shallow Neural Network (QCNN), can be equivalently trained via solving convex feasibility problems iteratively. With the quasiconvex structure, QCNN allows for efficient training with theoretical guarantees. We verify the proposed QCNN using several common machine learning tasks. The quasiconvex structure in QCNN demonstrates even better learning ability than non-convex deep networks in some tasks.

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
