# OpenReview forum: "Quasiconvex Shallow Neural Network"
_ICLR.cc/2023/Conference — Submitted to ICLR 2023_

### Official Review · Reviewer_kN6h · 2022-10-14

**Confidence:** 4
**Correctness:** 3
**Technical Novelty And Significance:** 3
**Empirical Novelty And Significance:** 3
**Recommendation:** 3

**Clarity, Quality, Novelty And Reproducibility:**

Clarity:
This paper is well-written in general. Just some questions for the experiments:
1. What are the exact architectures for deep networks and QCNN used in the experiments?
2. Is the QCNN trained by (stochastic) gradient descent or by solving the convexity feasibility problem using a non-local-search algorithm?

Quality:
The theory part is solid, but the experiments are restricted to small synthetic datasets or some easy tasks. The experiments would be far more convincing if QCNN can outperform deep networks in standard vision tasks, such as CIFAR-10 or ImageNet.

Novelty:
I think the idea of a QuasiConvex neural network is novel.

Reproducibility:
I think the theoretical analysis and the experiments are both reproducible.

**Strength And Weaknesses:**

Strengths:
1. I think it's important to either prove theoretical guarantees for the widely used deep neural networks or propose new architectures that admit simpler theoretical analysis. This paper made progress along the second direction by proposing QCNN that can be optimized efficiently with theoretical guarantees.
2. It's also interesting that QCNN outperforms deep neural networks on some tasks when the tasks involve some convexity characteristics. This might guide the design of neural networks for these types of tasks in the future.

Weaknesses:
1. This paper claimed that deep neural networks are likely to be trapped in bad local optimal with a large loss, which I doubt. Although bad local minima do exist, stochastic gradient descent can usually find good solutions on standard tasks with modern architectures. There has been much recent progress in explaining this surprising phenomenon for over-parameterized neural networks (notably the NTK theory and the mean-field theory).
2. Although over-parameterized deep networks may not be difficult to train empirically, I do agree that it's important to either prove training guarantees for these deep networks or propose other architectures that admit easier analysis. However, the proposed architecture in this paper (QCNN) is very limited in its representation power and may not be able to replace modern deep networks. Although QCNN can be stacked infinitely, as the authors mentioned in the paper, too many minimization pooling layers can damage the representation power. All the experiments showing the advantage of QCNN are restricted to synthetic datasets or easy tasks, which further reinforces my impression of the limitation of QCNN.



**Summary Of The Paper:**

1. This paper proposed the QuasiConvex shallow neural networks (QCNN) whose optimization is equivalent to a convex feasibility problem, which can be solved efficiently with theoretical guarantees. As the building block, they first proved that the multiplication of ReLU output is quasiconvex. These building blocks can then be integrated by minimization pooling layers while preserving the quasi-convexity.
2. In the experiments, this paper showed that QCNN outperformed deep neural networks in some tasks, in particular when the tasks involve some characteristics related to convexity.

**Summary Of The Review:**

I think it's certainly an important direction to either prove training guarantees for deep neural networks or propose alternative architecture with theoretical guarantees. Although the proposed QCNN enjoys training guarantees, its representation power seems to be very limited. So I am not sure to what extent QCNN can replace the modern deep networks. All the experiments showing the advantage of QCNN are restricted to synthetic datasets or easy tasks.

---

### Official Review · Reviewer_QKhx · 2022-10-22

**Confidence:** 3
**Correctness:** 2
**Technical Novelty And Significance:** 2
**Empirical Novelty And Significance:** 2
**Recommendation:** 5

**Clarity, Quality, Novelty And Reproducibility:**

Clarity:
As discussed above, several claims in the paper are not well established. Experimental section is unclear.

Quality:
Most of the theoretical parts are clear but there is an unclear step in the optimization algorithm.

Novelty:
The paper includes a novel idea (quasi-convex networks).

**Strength And Weaknesses:**

Strengths:
1. Novel neural architecture with formal optimization guarantees.
2. Introducing quasi-convex/concave structures to neural networks is interesting.

Weaknesses:
1. The paper only focuses on the regression setting, while it is written as if the results hold more generally (i.e., in both classification and regression settings). Furthermore, the quasi-convex network is shallow.
2. In the paper there are several claims which are not well established:
(a) It is claimed that neural networks can fail to optimize. The main example given to support this, is the theoretical work of Brady et al. (1989). They show that backprop fails in a regression setting with logistic activation functions. In contrast, there are many theory papers which show that for modern-day networks with ReLU activations, optimization is tractable for overparameterized neural networks (e.g., see papers on NTK). Thus, the claim that optimization is an issue for neural networks is not well established.
(b) It is claimed that “The non-convex network could fail, i.e., being trapped in bad local optima with large errors, especially when the task involves convexity (e.g., linearly separable classification)”. The claim that networks fail in tasks that involve convexity is not well established and seems to be only based on the example of Brady et al. (1989). As mentioned, the latter example is in a regression setting and networks with logistic activations (which, to my knowledge, are not widely used in practice). I am not aware of papers which show that networks with ReLU activations fail when classifying linearly separable datasets.

3. The training procedure is unclear. How is the network output restricted to be smaller than the network labels? There are no details on this.

4. Some of the experimental settings are unclear. (a) In the synthetic dataset, which neural network was used? It seems odd that a neural network cannot fit the curves exactly. What were the training points in these experiments? (b) The experimental setting of the change-point detection and contour detection are unclear. What exactly are the data and labels and what is the regression/ classification task?


**Summary Of The Paper:**

This work suggests a novel shallow neural network architecture that is quasi-concave. Consequently, the network can be trained in regression settings with convex optimization tools that have optimization guarantees. The efficacy of the network is demonstrated in several practical tasks.

**Summary Of The Review:**

Interesting architecture and idea of introducing quasiconvexity. However, the claims in the paper are not well supported and the experimental settings should be clarified. Furthermore, there is currently an unclear step in the optimization algorithm.

---

### Official Review · Reviewer_xAee · 2022-10-22

**Confidence:** 4
**Correctness:** 2
**Technical Novelty And Significance:** 2
**Empirical Novelty And Significance:** 2
**Recommendation:** 3

**Clarity, Quality, Novelty And Reproducibility:**

-Quality and Reproducibility: The theoretical proof about the quasiconcavity appears to be correct. However, since a lot of details about network structure, parameters and training are missing, many claims in this paper do not seem to be supported by enough evidence. Furthermore, it could be hard to reproduce the results in this paper due to the lack of details.

-Novelty: To the best of my knowledge, the idea of building a neural network using ReLU with min-pooling is novel.

-Clarity: This paper is generally well-structured. However, a lot of details are missing in this paper, e.g., the network, algorithm, and parameters used in the experiments. This makes many claims and conclusions in this paper unclear and less convincing. There are also a few minor typos/formatting issues, which are listed below:

1. Paragraph after "Issue 2", lines 3-4, "network structure can be convexity" -> "network structure can have convexity"
2. In-text citation format should be fixed for some places, i.e., the parenthesis should include both the author name and publication year. E.g., page 2, "Amos et al. (2017)", and section 6.2, the paragraph for "Change point detection of distributions".
3. For Figure 6(a), what is the contour plotted? Is it the ground truth or produced by QCNN/deep neural network?

**Strength And Weaknesses:**

-Strengths:

1. The idea of combining ReLU outputs and min-pooling to construct quasiconvex functions is interesting and novel. It provides a tradeoff between the strict convexity of some previous models and the high degree of non-convexity in commonly-used deep neural networks.

2. The authors include many figures in their paper, which helps with the understanding of their arguments, network design, and tasks in the experiments.

-Major Weaknesses:

1. Many essential details about the network training process and the experimental setting are missing in this paper, which makes the results and claims in this paper much less convincing. I will explain this in detail below:

1.1 For minimization pooling, how do the authors select the weights $a_1,\cdots,a_n$ in the weighted minimization pooling $\phi$?

1.2 When transferring the quasiconvex optimization problem into a feasibility problem, how do the authors choose the function $\varphi_t(\theta)$? Besides, how to make sure that the network output is always no larger than the labels?

1.3 In the experiment section, what are the network structures used for QCNN and "deep neural networks"? Is the same structure used for every task? What optimizer and hyperparameters are used for training the deep neural networks? It is difficult to draw conclusions about the performance of deep neural networks without knowing any such detail.

2. It is unclear how much advantage in representation power QCNN has over convex neural networks because the structure of QCNN appears to be quite restricted. Specifically, QCNN only contains linear layer, ReLU, and min-pooling and cannot do addition operations on intermediate representations of the networks, deeper QCNNs might not have better representation power than shallower ones. This paper doesn't provide a theoretical analysis of the representation power of QCNN, and the experiments section doesn't provide the performances of convex neural networks.

3. The scale of the experiments is somewhat small, and it seems the authors did not run the experiments multiple times. These makes the experimental results less convincing.

4. This paper doesn't provide theoretical guarantee for the convergence speed of the proposed network with the training algorithm, making it unclear whether QCNN is more efficient than previous methods.

**Summary Of The Paper:**

This paper proposes a new neural network architecture, i.e., QuasiConvex Neural Network (QCNN), which could make the training process quasiconvex with proper loss functions. QCNN is a stack of its basic building block, which is a linear layer with ReLU activations and min-pooling. The training process of QCNN was formalized as a convex feasibility problem and can be solved using existing algorithms, e.g., bisection. The authors also provide small-scale experimental results of shallow QCNN versus regular neural networks on synthetic data and several other datasets and show that QCNN is better in accuracies or convergence speed for some tasks.

**Summary Of The Review:**

I tend to vote for rejecting this paper. Although the idea of constructing a quasiconvex neural network in this particular way is interesting, many important details about the network and especially the experiments are missing, making the claims less convincing. On the theoretical side, this paper doesn't provide enough theoretical guarantees about the representation power and training efficiency of the proposed network.

---

### Official Review · Reviewer_B2PF · 2022-10-24

**Confidence:** 4
**Correctness:** 3
**Technical Novelty And Significance:** 3
**Empirical Novelty And Significance:** 2
**Recommendation:** 3

**Clarity, Quality, Novelty And Reproducibility:**

The writing is generally of good quality, although (as mentioned above) some of the mathematical developments are imprecise or unclear. The quality of referencing could be improved, as sometimes the wrong citation is used for classical works. See "Minor Comments" above.

I have not seen any previous works which attempt to derive quasi-concave neural networks, so the submission is novel as far as I am aware.
However, it is not reproducible. Many details required to reproduce the experiments results are omitted (step-size, batch-size, optimizer, etc). For instance, it is not even stated what procedure is used to solve the convex feasibility problems necessary for training the quasi-concave models.

**Strength And Weaknesses:**

The main strength of this submission is the simplicity and elegance of the construction for quasi-concave neural networks.
Endowing neural networks with additional structure that makes them possible to globally optimize while preserving some of their
representation power is a great approach to principled deep learning and valuable for the ICLR community.
The manuscript text is also polished and contains many nice figures which contribute to an intuitive understanding of the quasi-concave architecture.

However, I feel the strengths of this submission are outweighed by its weakness, which are summarized in the following points:

- Important limitations of the quasi-convex architecture are not addressed in the main text. The proposed architecture can only represent non-negative functions, which is a significant weakness for regression problems. However, this is completed elided and could be missed by the casual reader.

- The submission is not always rigorous and some of the mathematical developments are unclear. For example, see the development of the feasibility algorithm in Eq. 4 and Eq. 5. Firstly, $t \in \mathbb{R}$ while $y, f(\theta) \in \mathbb{R}^n$, where $n$ is the size of the training set, so that the operation $y - t - f(\theta)$ is not well-defined. Moreover, even if $y, f(\theta) \in \mathbb{R}$, the inequality $\psi_t(\theta) \leq 0$ implies $l(\theta) \leq t^2 / 2$, rather than $\(\theta) \leq t$. Since, in general, the training problem will be defined for $y \in \mathbb{R}^n$, the derivations in the text should handle this general case.

- The experiments are fairly weak and do not convince me that the proposed models have sufficient representation power to merit use over kernel methods and other easy-to-train models. The main issue here is the experimental evaluation does not contain a single standard benchmark problem nor does it compare against standard baseline methods. For example, I would really have liked to see regression experiments on several UCI datsets with comparisons against kernel regression, two-layer ReLU networks, etc. Although boring, such experiments establish a baseline capacity for the quasi-concave networks; this is necessary to show they are "reasonable". The experiments as given have several notable flaws:

    - Synthetic dataset: This is a cute synthetic problem, but obviously plays to the strength of the quasi-concave models. I would have preferred to see a synthetic problem for which was noisy with non piece-wise linear relationship.
    - Contour Detection Dataset: It is standard to report the overall test ODS, instead of reporting it on different subgroups. This allows the reader to make a fair _overall_ comparison between the two methods.
    - Mass-Damper System Datasets: This is a noiseless linear regression problem in disguise, so it's not surprising that quasi-concave networks perform well.
    - Change-point Detection: Again, I would really have rather seen some basic benchmarks like MNIST before moving on to novel applications like detecting changes in data distribution.

#### Minor Comments

**Introduction**:
    - The correct reference for SGD is the seminal paper by Robbins and Monro [1].
    - The correct reference for backpropagation is Rumelhart et al. [2]
    - "Issue 1: Is non-convex deep neural networks always better?": "is" should be "are".
    - "While some experiments show that certain local optima are equivalent and yield similar learning performance" -- this should be supported by a reference.
    - "However, the derivation of strong duality in the literature requires the planted model assumption" --- what do you mean by "planted model assumption"? The only necessary assumption for these works is that the shallow network is sufficiently wide.

**Section 4**:
    - "In fact, suppose there are m weights, constraining all the weights to be non-negative will result in only $1/2^m$ representation power." -- A statement like this only makes sense under some definition of "representation power". For example, it is not obvious how non-negativity constraints affect the underlying hypothesis class (aside from forcing it to contain only non-negative functions), which is the natural notion of representation power.
    - Equation 3: There are several important aspects of this model which should be mentioned explicitly in the text. Firstly, it consists of only one neuron; this is obvious from the notation, but should be stated as well. Secondly, it can only model non-negative functions. This is a strong restriction and should be discussed somewhere.
    - "Among these operations, we choose the minimization procedure because it is easy to apply and has a simple gradient." --- the minimization operator may produce a non-smooth function, which does not admit a gradient everywhere. Nor is it guaranteed to have a subgradient since the negative function only quasi-convex, rather than convex.
    - "... too many minimization pooling layers will damage the representation power of the neural network" --- why? Can the authors expand on this observation?

**Section 5**:
    - "... if we restrict the network output to be smaller than the network labels, i.e., $f(\theta) ≤ y$" --- note that this observation requires $y \geq 0$, which does not appear to be explicitly mentioned.
    - What method is being used to solve the convex feasibility problem in Eq. (5)? I cannot find this stated anywhere.

**Figure 6**:
    - Panel (b): "conveyers" -> "converges".

**Figure 7**:
    - The text inside the figure and the labels are too small to read without zooming. This text should be roughly the same size as the manuscript text.
    - "It could explain that the classification accuracy of QCNN (94.2%) outperforms that of deep networks (92.7%)" --- Is this test accuracy, or training accuracy? I assume this is the test metric on the hold-out set, but the text should state this clearly.
### References

[1] Robbins, Herbert, and Sutton Monro. "A stochastic approximation method." The annals of mathematical statistics (1951): 400-407.

[2] Rumelhart, David E., Geoffrey E. Hinton, and Ronald J. Williams. "Learning representations by back-propagating errors." nature 323.6088 (1986): 533-536.

**Summary Of The Paper:**

This submission develops neural network architectures for which the squared-error training objective is quasi-c.
The proposed models are based on the observation that composition of single ReLU neurons preserves quasi-concavity, as does
the minimum-pooling operation.
By combining these operators, the authors obtain quasi-concave neural networks capable of representing
non-negative piece-wise linear functions.
They propose a training algorithm based on bisection search that solves convex feasibility problems as a sub-routine.
The submission concludes with empirical evaluations on both synthetic and real datasets.

**Summary Of The Review:**

This is an interesting attempt to develop a model class somewhere between the representation power of non-convex neural networks
and the polynomial-time optimization guarantees of convex models. While the quasi-concave neural networks
are elegant and mathematically appealing, their utility is not effectively demonstrated by the experimental evaluation and a more rigorous presentation of the training algorithm is necessary.
I think this could be a strong submission if sufficient baseline experiments are added and I encourage the authors to continue the work.
However, I feel that it is not ready for publication at this time.

---

### Decision · Program_Chairs · 2023-01-20

**Decision:**

Reject

**Justification For Why Not Higher Score:**

All reviewers agreed that the paper is a clear reject (see weaknesses above).

**Justification For Why Not Lower Score:**

N/A

**Metareview: Summary, Strengths And Weaknesses:**

Strengths:
- the idea of quasiconvex networks is interesting
- the paper is well-written and contains some illustrations

Weaknesses:
- the proposed network is shallow
- many claims are not sufficiently backed up with evidence
- the training procedure and experimental setup are unclear
- the network can only model non-negative functions
- the advantage over usual neural networks is unclear

Overall, there is agreement among the reviewers that the paper, while based on an interesting idea, is not ready for publication yet.

The authors didn't write a rebuttal.